# Animal Protection in Filming in the Context of Spain

**DOI:** 10.3390/ani13071144

**Published:** 2023-03-24

**Authors:** Pablo De Damborenea Martín, Rubén Bermejo-Poza, Jesús De la Fuente Vázquez

**Affiliations:** Animal Production Department, Veterinary Faculty, Complutense University of Madrid (UCM), Avenida Puerta de Hierro s/n, 28040 Madrid, Spain; pdmartin@ucm.es (P.D.D.M.); rbermejo@ucm.es (R.B.-P.)

**Keywords:** animal welfare, animal protection, European legislation, Spanish regulations, filming

## Abstract

**Simple Summary:**

The history of cinema could not be understood without the participation of animals. Since the birth of filmmaking (the seventh art), these main protagonists have made it possible to faithfully recreate reality, not only forming part of the film set, but also playing roles that have had a great influence and impact on society at different times. However, unlike what has happened in other sectors, in which the development of animal protection regulations and welfare assessment systems have experienced tremendous growth since the mid-20th century, the mention of this type of activity is nearly residual in the legislation currently in force. Taking into account the fact that this industry has a significant economic impact, that there is a growing social demand related to obtaining products that are respectful of animals, and that the Treaty on the Functioning of the European Union urges countries to take into account all issues related to animal welfare in the development of its policies, it is necessary to highlight the legal vacuum that exists today in the animal-related regulations of the different countries of the European Union.

**Abstract:**

Animals have fundamentally contributed to the development and growth of different cinematographic and audiovisual activities. As established in article 13 of the Treaty on the Functioning of the European Union (TFEU), the basic text that establishes the objectives of the EU and from which the regulations or directives emanate, animal welfare is protected for the good of the general interest and therefore, must be taken into account when developing different European Union policies, especially those that may affect the internal market. This work aims to analyze each of the European regulations on animal protection and welfare to subsequently focus on each of the animal protection codes of the different countries of the union, trying to determine those premises related to the protection or the welfare of the animals that participate in these types of activities. We also wanted to make a brief comparison with countries such as the United Kingdom (UK) or the United States of America (USA), which are highly relevant to this topic. As a result of this analysis, the absence of standardized norms in the European Union has been revealed; however, in the UK and the USA, specific laws related to animal use in film media exist. Therefore, to develop new standards that guarantee the protection of animals in audiovisual productions in the EU are necessary, and these standards could also be expanded to other sectors or activities related to working animals.

## 1. Introduction

Animals alongside humans since prehistoric times [1]. Humans have used animals as a source of food or work, and as a key element offering different alternatives or means of entertainment to the population [2]. In the specific case of the cinema, the use of animals is as old as filmmaking itself (the seventh art). The first example of what could be considered a cinematographic sequence dates back to the 19th century, when Edward James Muggeridge, known by the pseudonym Eadweard Muybridge [3], managed to create the illusion of the movement of a horse by joining different frames obtained during a race [4]. Subsequently, the Lumière brothers screened what could be considered the first film in history, in which, unintentionally, the importance of animals at the time was revealed, showing in their images some of the elements that formed a fundamental part of the props of most of the scenarios of the time, such as horse-drawn carriages or stray dogs. Since then, many animals have appeared in film, some with historically important roles [5]. Examples include the first films set in the Wild West, with the participation of hundreds of horses; Rin Tin Tin or Lassie dogs; the Flipper dolphin; the Cheetah monkey; the King Kong gorilla; the ravens used by Hitchcock; and even the famous lion that appears at the beginning of all the films made by the Metro–Goldwyn–Mayer production company. The history of cinema would not be the same without the presence of these protagonists [6,7]. Even in the digital age, where great advances in technology could minimize the use of live animals [8], they continue to play a highly relevant role in the production of films and advertising campaigns throughout the world [9].

In Spain, an example of this can be extracted from the data provided by the Madrid City Council, which keeps a record of the participation of animals in this kind of activity. Since the passing of the Animal Protection Law 4/2016 [10], between 2017 and 2020, a total of 786 requests were made to use animals in events, including films. Another important piece of data, which has been provided by one of the main companies dedicated to and specializing in audio-visual productions with animals in Spain, reveals that this entity alone completed around 374 shoots during this same period. In both cases, it is difficult to know the exact total number of animals that participated in these activities. However, considering that at least one animal was used in each shoot, the total volume of individuals is not negligible.

Moreover, it is necessary to consider the economic impact of this kind of activity. Focusing on the European market, the film sector alone raised more than EUR 12,000 million between 2019 and 2021, according to data from the International Union of Cinemas (UNIC) [11]. In Spain, around EUR 302 million from General State Budgets were earmarked for these activities between 2018 and 2021 [12], with Spanish films generating approximately EUR 283 million through ticket sales alone [13].

On the other hand, the participation or use of animals in filming has always been surrounded by criticism or complaints by different organizations. In this sense, the People for the Ethical Treatment of Animals (PETA) has repeatedly denounced situations of mistreatment and even the death of some animals during the filming of major film productions, such as *The Hobbit* (2012), *Life of Pi* (2012) [14], or the film *A Dog’s Purpose* (2017), in which a video was leaked in which it could be seen that a dog was about to drown during the recording of an action scene [15]. In Spanish cinema, it is also possible to find some complaints, such as the one filed against the film *Talk to Her*, directed by Pedro Almodóvar, which was involved in a great controversy due to the use and slaughter of several bulls during filming [16], or the movie *Blancanieves* (Snow White), directed by Pablo Berger (2012), for the same reason, and in which the High Court of Justice of Madrid forced the Community of Madrid to open a sanctions proceedings for the death of nine young bulls during the filming of the movie. However, despite its relevance, as has been pointed out, today there is no regulation at the European level that guarantees the protection or welfare of animals in this kind of economic activity [17]. Therefore, the creation of a specific regulatory framework for this type of activity or its inclusion in the existing regulations is a pending task.

## 2. Evolution of Legislation and the Current Situation

In order to highlight the lack of regulations, we carried out a detailed study of all European Union (EU) regulations, as well as the animal protection codes of each country in the EU (Table 1). Access to all European legislation is available in different formats, mainly through the European Union’s website [18], with free access for any citizen. In the case of the legislation of each country, it is necessary to carry out a specific search through the official government pages. Finally, all the animal protection regulations included in the Spanish legislation are found in the Boletin Oficial del Estado (BOE) [19].

At the European level, in 1957, article 36 of the Treaty of Rome [20], the first founding treaty of the European Economic Community (EEC), provided that member states could establish import, export, or transit restrictions in order to protect the health and lives of animals, which could be considered the first approach to issues related to animal welfare. However, it was not until 1974 that an attempt was made to regulate operations related to animal slaughter [21]. At this point, it is important to understand the social context in which Europe found itself from the late 1940s. After overcoming two periods of great war, the priority was to feed the population, leaving aside the means used to guarantee this supply, thus placing animals in an unfavorable position [22]. This reality was made clear in the conclusions gathered at the end of the 20th century in the famous Brambell Report [23], commissioned by the British government after the publication of the book *Animal Machines,* written by Ruth Harrison, in 1964 [24], which was critical in the development of future legislation and protocols for the evaluation of animal welfare.

From that moment on, many important developments occurred, leading to the treaties of Amsterdam in 1997 and Lisbon in 2007, which recognized animal sentience [25]. Today, the European Union could be considered to have one of the highest animal protection standards in the world [21]. In this sense, the large areas encompassed by the existing legislation (to date) are collected in Table 1 [26].

Table 1 contains the main animal protection regulations in force in the European Union. However, there are many other specific regulations that are not included due to their extension, but these can be accessed in the database available on the official website of the EU and EFSA [27]. Among these legal texts, it is necessary to establish a primary distinction between regulations, which are legal acts that must be applied automatically and uniformly in all EU countries, and directives, whose objective is to impose the achievement of a certain objective, while giving the individual countries the freedom to choose the necessary tools to achieve it [28].

**Table 1 animals-13-01144-t001:** Main European regulations on animal protection and welfare.

Regulation	Area
Directive 98/58/EC [29]	Animal husbandry
Regulation (CE) No. 1/2005 [30]	Transport
Regulation (CE) No. 1099/2009 [31]	Slaughter
Regulation (EC) No. 407/2009 [32]	Conservation
Directive 92/43/CE [33]	Fauna
Directive 2010/63/EU [34]	Experimental animals
Directive 1999/22/CE [35]	Zoos
Regulation (CE) No. 998/2003 [36]	Non-commercial movements of pet animals

Finally, it is necessary to include in this list the European Convention for the Protection of Pet Animals, presented in Strasbourg on November 13, 1987 [37], as well as the Universal Declaration of Animal Rights [38]. This last text, despite not having legal validity, contains points that are included in the legislation of different countries [39], and therefore, it is relevant to point them out.

## 3. Animal Protection in Filming

### 3.1. European Regulatory Framework

First of all, it should be noted that none of the community regulations indicated in Table 1 are related to the participation of animals during filming. However, most of the premises included in these directives could be extrapolated or adapted in a text that regulates the development of this kind of activity.

The Council of Europe was one of the first international institutions to initiate discussions on animal welfare in Europe. It should be noted that the Council of Europe has played a decisive role in defining the legislation of the European Economic Community initially and of the current European Union, since a large portion of the European countries belong to both institutions. In this sense, in relation to the protection of animals in filming, the Council of Europe passed legislation through the European Convention for the Protection of Pet Animals of 1987 [37]. References to the participation of animals in this type of activity can be found in two articles, 4 and 9, in which the person responsible for filming must take into account the needs of the animals and provide them with adequate care, preventing their health and well-being from being endangered. It is noteworthy that the text only makes explicit reference to advertising, so the rest of the audio-visual activities, such as the cinema, are supposed to be included in what is referred to as “similar manifestations”. This fact could lead to an ambiguous and impractical interpretation. Moreover, there is also mention of the conditions that the organizer of the activity must create; however, these are very general and do not consider many of the questions that would be necessary to guarantee animal welfare. Although the member states are not legally obliged to sign this document, they may be expected to do so, since according to Council of Europe’s Statute, each State undertakes to “collaborate sincerely and effectively in the achievement of the objective of the Council.” In this sense, to date, the convention has been ratified by 26 Council countries [40].

Instead, the Universal Declaration of Animal Rights, a text that is not legally binding, refers, in article 13b, to violent scenes, considering that they should be banned unless their goal is to denounce animal mistreatment. In addition, article 7 reflects the idea that the work carried out by any animal must be limited in time and effort, which could be extrapolated to audio-visual productions. In the case of time limitation, an issue that undoubtedly takes priority in work legislation for human beings and especially for minors, the proposal could be very successful. However, its consideration should be supported and argued not only based on theoretical criteria, but also on practical results of evaluations carried out while animals are working. The main drawback of these articles is that they remain recommendations subject to the subjective interpretation of those responsible for the activity. Although the aim is commendable, in practice, it may not be useful, since it lacks legal validity, as previously mentioned. Regarding issues related to scenes that simulate violence or mistreatment, in many cases, another drawback arises: the lack of an inspection agency or institution to certify that the techniques used to carry out certain actions are adequate and are not harmful to animals.

In all European Union countries, there is legislation regarding animal protection; however, not all examples refer to the use of animals in filming. In those countries that do include filming, the only reference is in relation to the scenes that simulate violence and/or mistreatment of animals. Table 2 shows that of the 27 countries that currently make up the EU, only 12 of them make specific mention of this issue. It is worth noting the case of Slovenia [41], a country that establishes the requirements in a more developed way than any other.

In some countries, such as Italy [52], Ukraine [53], Holland [54], Estonia [55], or Norway [56], no direct reference is made to filming, although in their legislation, it is possible to find terms such as “exhibitions,” or “entertainment activities with animals,” which could be included in filming. Another example might be the Latvian Animal Protection Law [57], which specifically defines the terms “exhibit animal,” as referring to those kept for public display for public entertainment or public education, and “work animal,” considered as one that has acquired specific skills and performs actions specified by a human. Without a doubt, it would be useful and necessary to carry out a more exhaustive and detailed study of each of these laws in order to determine whether in practice there is any other provision that should be taken into account when carrying out an activity of this type.

### 3.2. Regulatory Framework in Spain

At a territorial level, Spain is one of the most decentralized states in the world, and this affects animal protection policies, whose development is a responsibility of the autonomous communities. As for the specific animal protection regulations, Spanish legislation contains different laws, some of which derive directly from guidelines and decisions adopted by the European Parliament. Those standards are all collected in the Spanish Official Bulletins, referred to as BOE [58]. As with European legislation, there are other regulations that complement the main examples, and therefore, have, although indirectly, an impact on animal welfare (Table 3). The latter will not be referred to here to avoid repetition, since they can be accessed in the BOE.

Spain has hosted the largest number of film productions since the mid-20th century, including famous Westerns. However, it is striking that, despite the fact that it has specific regulations for cinematographic activities and audiovisual arts, there is no specific section related to the participation of animals.

The Organic Law 10/1995, of the November 23 Penal Code [67], establishes the sanctions related to the crimes of abandonment and mistreatment committed against animals. It could be included in Table 3; however, it only protects what are considered domestic or tamed animals, leaving out wild fauna. This has brought about numerous criticisms, including the establishment of categories or privileges within the animal world due to different degrees of protection. It is particularly important in the case of filming in natural environments, since there is no specific law for filming animals in natural spaces. This produces a legal vacuum when establishing the limits on the actions that can be carried out.

In September 2020, a proposal was made to modify the civil code in Spain. In order to modify the mortgage law and Law 1/2000, of January 7, in regards to civil prosecution [68] and the legal regime of animals, through which, animals would cease to have the status of movable property in the Spanish legal system, and would be considered “living beings endowed with sensibility.” However, it could be considered that this reform comes too late if we compare it with other European countries, such as Austria (since 1986) or Germany (since 1990), who already recognized it in their civil codes, and who even include it as one more point in their constitutions, unlike the Spanish code, which makes no mention of animals in a general or specific way.

Despite the extensive regulatory framework included in Table 3, there is no legislation that refers directly to issues related to the use of animals in work activities and specifically to filming, despite its enormous historical importance and economic impact. However, the application and many of the premises contained in these texts could perfectly be extrapolated and adapted to a specific text that regulates work activities, such as filming. Only Law 32/2007 on the care of animals in exploitation, transport, experimentation, and slaughter, considers it a very serious infraction “to use animals in film, television, artistic or advertising productions, even with the permission of the competent authority, when their death occurs.”

In the absence of a framework or general law on animal welfare in Spain, the autonomous communities, and even the municipalities, have the capacity to legislate on this matter. This has meant that each autonomous community, and even each municipality, has drawn up their own laws. These laws mean that the legal requirements in each autonomous community or each municipality may be different and may directly affect the development of a film, since there may be differences between neighboring municipalities, which makes it difficult to establish restrictions (Table 4).

Due to the breadth of the subject, it is not possible to include regional or municipal legislation in this review, although some may be cited later, if they mention the protection of animals during filming.

The only specific reference to filming included in these regulations has to do with scenes that simulate possible animal abuse. In some cases, such as Law 11/2003 of November 24 [69] on the protection of animals of the Autonomous Community of Andalusia, article 4, it is prohibited to “use animals in exhibitions, circuses, publicity, advertising, popular festivals, and other activities if this causes suffering, pain or unnatural treatment for the animal.” Another example would be the Law for the Protection of Animals in Catalonia, (Decreto Legislativo 2/2008) [75], which prohibits the use of protected species, such as the brown bear, all cetaceans, or lynxes, in commercial activities. However, just as in the European regulations, the terms “exhibition,” “advertising,” or “commercial activity” are not sufficiently defined and, therefore, their interpretation and application can vary.

The communities of Aragón, the Principality of Asturias, Cantabria, and Castilla-La Mancha do not even make a direct reference to this type of activity. It is also necessary to point out that the municipalities of each community can also introduce the modifications they deem appropriate, as long as it makes the codes more restrictive. This lack of common criteria may mean that the requirements demanded in each locality are different, which causes an added difficulty when properly planning activities of this type, taking into account the particularities of each zone. It is worth highlighting the modifications introduced in Law 4/2016 of the Community of Madrid [10], one of the most modern, which includes the need to request authorization from the municipalities where animals are to be filmed. This authorization would be linked to compliance with a series of requirements related to the health status of the animal, or to compliance with the provisions of the standard itself with respect to identification and preventive treatments. However, the competence in the application of this law is transferred again to each city council, which can translate into differences in criteria, both at the level of documentary requirements and in terms of deadlines, creating confusion that affects the effectiveness of the standard and is a problem for those responsible for the filming, who often choose not to comply with it.

## 4. Animal Protection in Filming in the United States and the United Kingdom

It is pertinent to review animal welfare legislation in other countries outside of Europe known for their long tradition in the protection of animals, specifically, the United States of America (USA) and the United Kingdom (UK), to determine how the participation of animals in movies, television, or publicity is regulated. It is necessary to point out that the legal system of these two countries, in relation to their legislative competencies or the division of powers, are different from each other, and also from Spain, and this has consequences regarding the type of regulations that exist in each of these countries.

In the USA the Animal Welfare Act (AWA) [79], which dates from 1966, is the only federal law in the United States that specifically regulates how animals should be treated in aspects such as research, transport or trade, and the participation of animals in public exhibitions. There are also agencies such as the United States Department of Agriculture (USDA), the Animal and Plant Health Inspection Service (APHIS), and Animal Care, that are responsible for executing and controlling its compliance [80]. A notable aspect of this law is that it requires that the owners of the animals that are going to work in these audiovisual media and those defined as “exhibitors” are in possession of the permits that the USDA requires. In addition, people who work in the handling of animals (animal handlers) must also be in possession of licenses granted by this department for the development of these activities.

In the United Kingdom, there is a general animal welfare law (Animal Welfare Act, 2006) [81] developed in 2006 by the Department of Environment, Food, and Rural Affairs by the modification of other older legislation. This law includes the need to have a license or registration that must be provided by a local or national authority to carry out certain activities in which animals are under the responsibility of any person involved. In the UK, we find other more specific laws related to the participation of animals in films, such as the Cinematograph Films (Animals) Act 1937 [82] and the Performing Animals (Regulation) Act 1925 [83], which prohibits the exhibition or distribution of films in which animals may have been mistreated or treated cruelly.

Moreover, along with the aforementioned legislation, in both countries, we find the existence of two organizations that are in charge of advising companies in the sector, establishing, based on the legislation of each country, guidelines that serve as a guide for good practices in working with animals.

The United Kingdom, in addition to producing many of the greatest thinkers and scientists who helped to improve the situation of animals and their status in society, produced one of the first laws to prohibit “the cruel and improper treatment of cattle” in 1822. Two years later, in 1824, the “Royal Society for the Prevention of Cruelty to Animals” (RSPCA) was born. It can be considered the first and oldest society to prevent cruelty to animals, which would extend in later years, reaching the USA, Australia, and New Zealand. This organization, among many of the functions that it performs in favor of the protection of certain animals, has developed a protocol or guide of best practices [84] that, without being an official document, helps with the goal of guiding and advising companies in the production sector on how to handle animals in a way that guarantees their welfare.

In the USA, there is an association called American Humane (AH), which has also prepared a document as a guide in which animal welfare standards are established that are higher than the anticruelty laws that apply to all the states and whose objective is to regulate the participation of animals in filming in order to ensure their welfare and protection [85]. To this end, many of the guidelines they include are standards established by common sense, but they are also drafted based on federal, state, or local legislation. In addition, in this guide, the American Humane Association urges companies to request information from the USDA about the owners of the animals, as well as about the facilities where they keep or train them, in order to reject providers who have recent and/or repeated incidents of animal abuse or neglect or other USDA violations related to the care and treatment of animals.

## 5. Conclusions

The social awareness that currently exists regarding the care and protection of animals is greater than at any other time. This fact has promoted the development of laws in the European Union and different countries. However, there are currently certain areas that have not yet been legislated, such as work activities in which animals participate, including filming.

Although the history of the cinema could not be understood without the presence of these actors, there is no European standard that determines the requirements that must be met in relation to their participation. Currently, only issues related to scenes that simulate abuse are included; however, other necessary aspects that are essential to guarantee a high degree of animals protection are not taken into account. This is despite the fact that most of the regulations include premises that could be adapted in a text that specifically regulates this type of activity.

The Treaty on the Functioning of the European Union, TFEU, in its article 13 establishes that when formulating and implementing certain Union Policies, animal welfare must be taken into account. With this mention of animal welfare and also, since article 13 is within the title of provisions for general application, it has been considered that the European Union can regulate new issues on the welfare of other animals that are not within the policies of European Union. However, some answers given by the European Commission mention that guaranteeing animal welfare is not an objective of the current treaties, and article 13 of TFEU does not constitute in itself a legal basis for legislation of the European Union. However, the Court of Justice of the EU established that the protection of animal welfare constitutes a legitimate objective of general interest for all countries. Therefore, taking all of these examples into account, it could be concluded that there would be a legal basis that justifies and supports the development of new animal protection laws related to other activities outside the common agricultural policy (CAP), such as those related to the audiovisual sector, which generates a great economic impact, and therefore, the welfare of animals that participate in this type of activity should be regulated.

## Figures and Tables

**Table 2 animals-13-01144-t002:** Sections of animal welfare acts or codes that mention filming by country (EU).

Country	Act	Section
Austria	Federal Act on the Protection of Animals, 1 January 2005 [42]	Chapter 1, Article 5.8.
Belgium	Decree relating to the Walloon Animal Welfare Code, 12 December 2018 [43]	Chapter III, section 1, Subsection 1, Articles D23 and D 24
Bulgaria	Animal Protection Act, 31 January 2008 [44]	Chapter III, Article 28
Finland	Animal Welfare Act, 4 April 1996 [45]	Chapter II, Section 19
Germany	Animal Welfare Act, 18 May 2006 [46]	Article 3.6
Greece	Animal Protection Law, 2 February 2012 [47]	Articles 7 and 16
Lithuania	Law on Welfare and Protection of Animals, 6 November 1997 [48]	Articles 4.21 and 19
Luxembourg	Law on the Protection of Animal Life, Security, and Welfare, 6 June 2018 [49]	Chapter 3, Article 6.8
Portugal	Protection of Animals Act, 12 September 1995 [50]	Chapter I, Article 1.E
Slovenia	Act on Amendments to the Animal Protection Act, 24 April 2013 [41]	Article 14. a
Sweden	Animal Welfare Act, 20 June 2018 [51]	Chapter III, Article 1.3

**Table 3 animals-13-01144-t003:** Spanish regulations on animal protection and welfare.

Regulation	Area
Ley 8/2003 [59]	Animal health
Ley 32/2007 [60]	Care of animals during farming, transport, experimentation, and slaughter
Ley 33/2015 [61]	Natural heritage and biodiversity
Ley 31/2003 [62]	Zoos
Real Decreto 990/2022 [63]	Transport
Real Decreto 37/2014 [64]	Protection of animals at the time of slaughter
Real Decreto 630/2013 [65]	Spanish catalog of invasive alien species
Real Decreto 53/2013 [66]	Animals used for scientific purposes

**Table 4 animals-13-01144-t004:** Articles of the Animal Protection Code that mention filming by autonomous communities (Spain).

Autonomous Community	Regulation
Andalusia	Ley 11/2003. Article 5 [69]
Autonomous Community of Navarre	Ley Foral 7/1994. Title I, Article 6 [70]
Balearic Islands	Ley 1/1992. Title I, Article 4. 1c [71]
Basque Country	Ley 9/2022. Title I, Article 4. 5 [72]
Canary Islands	Ley 8/1991. Chapter I, Article 7 [73]
Castile and León	Ley 5/1997. Title I, Article 7 [74]
Catalonia	Decreto Legislativo 2/2008. Chapter II, Article 10 [75]
Community of Madrid	Ley 4/2016. Title II, Article 7ñ, 7o and Title VII, Article 19 [10]
Extremadura	Ley 5/2002. Article 4. 1c [76]
La Rioja	Ley 6/2018. Title II, Article 7 and Title IV, Articles 13a and 15 [77]
Valencian Community	Ley 4/1994 Title I, Article 7 [78]

## Data Availability

Not applicable.

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
