# Peer review of "Animal Protection in Filming in the Context of Spain"

_animals, 2023, doi:10.3390/ani13071144_

Round 1

Reviewer 1 Report

This is a very interesting manuscript which readers will find useful.

 Line 29 – perhaps it could be clearer whether the absence of standards is found in the UK, USA, EU (or all of them?)

Line 36 – to whom does “their” refer –  perhaps instead, say something along the lines that that animals have coexisted with humans from prehistoric times – you can use this is a reference  Linda Kalof, Looking at Animals in Human History, Reaktion Books, London (2007) 8-10.

Lines 9 and 39 – what is the seventh art.   Sorry for showing my ignorance!  I googled it and found out. There might be other readers who are also not aware but who are very interested in the subject matter of this article.

Lines 89-90 refer to “some European laws and treaties”  - would be helpful to cite a few of these.

Line 95 – is this study summarised in Table 1 – perhaps let the reader know.

Lines 97-101 – this is up to the editors – but the paragraph refers to websites and it would be helpful to the reader if these were cited.

Line 104 – “animal sacrifice” – could this be clarified. To English readers this invariably means some type of religious ritual where animals are killed.

Line 105 “at that time” is a little unclear because previous sentences refer to 1957 and 1974 – perhaps it could be replaced with “from the late 1940s” – or something similar.

Line 115 about the EU and its animal standards – I agree but it is such a broad statement a reference would be useful.   Eg https://www.fondation-droit-animal.org/proceedings-aw/the-european-union-legislation-on-animal-welfare/

Line 149 – perhaps point out that this declaration is not binding.

Line 159 – is there a reference for the conclusion that the article is rarely used – or is this conclusion drawn from the researchers’ own work.

Line 164 “each of the EU countries” – appears that not all EU countries have specific laws on filming – perhaps this wording can be amended as it appears to say that all EU countries have laws with respect to filming (made clear in line 166 onwards).

Line 178 – would be helpful to cite the Latvian law https://likumi.lv/ta/en/en/id/14940-animal-protection-law

Lines 206-7 – interesting comment regarding wild fauna.  Is this omission, however deliberate so that “pest” wild fauna are not subject to the law -  are there “environmental” type laws that would cover filming in natural spaces?  In New South Wales eg we have the Filming Approval Act 2004 (NSW) http://www.austlii.edu.au/cgi-bin/viewdb/au/legis/nsw/consol_act/faa2004177/  -   this is not designed to protect animal wellbeing but to protect the environment, which includes native animals, insects, etc.      As an aside we also have a code of practice specifically dealing with animals used in films (the code is mandatory under the Prevention of Cruelty to Animals Act 1979 (NSW) ) https://www.dpi.nsw.gov.au/animals-and-livestock/animal-welfare/animal-care-and-welfare/livestock/animal-use/animals-in-film-and-theatrical-performances/animals-in-film-and-theatrical-performances/code-of-practice-for-the-welfare-of-animals-in-films-and-theatrical-performances-plain-english

Line 213 – very interesting!

Line 230 – what is meant by “autonomous communities” – do you mean different stakeholder groups or local government (not sure)

Line 239 “extension” – do you mean given the breadth of the subject

Line 276 onwards – this part of the analysis requires modification – need to explain that USA and UK have different political systems which means they have different ways of dividing  legislative powers and competencies.

Lines 282-and generally that parag. USA has a similar federal system to Australia. Animal welfare laws are largely left up to the state legislatures and the federal government only becomes involved when it is considered a federal matter eg international trade, border controls for movement of animals and their products for biosecurity reasons.

There are no federal laws in USA dealing with films and animals – guidelines exist – I do not know I these are binding  (OK you refer to thse from line 318 onwards)

https://www.americanhumane.org/app/uploads/2016/08/Guidelines2015-WEB-Revised-110315-1.pdf

Also have a look at this https://www.animallaw.info/article/overview-laws-concerning-animals-film-media#:~:text=In%20response%2C%20a%20select%20few,a%20film%20depicting%20animal%20cruelty.   “Unfortunately, such anti-cruelty laws are ineffective against those depicting animal cruelty in films since it is difficult to identify the individual in the film. In response, a select few states specifically criminalize the filming of animal cruelty including California, Illinois, and Maine. These laws prohibit individuals from knowingly creating, selling, marketing, possessing a film depicting animal cruelty.

The UK has general laws that can apply to Britain, Wales and Northern Ireland. Scotland has some legislative powers devolved on it.

See also Wild Animals in Circuses Act 2019 – for exhibited animals https://www.legislation.gov.uk/ukpga/2019/24/contents/enacted   (does not apply to all animals)

 Lines 308-309 referring to anti-cruelty laws and RSPCA – needs modification.  The 1822 Act applied to farm animals such as cattle – not all animals eg it applied to horses, cattle and sheep and did not deal with welfare in the sense the term evolved in the second part of the 20th century. Rather, it deal with what can be described as abject cruelty.

The following article contains material that details how the failure to implement the 1822 Act led to the creation of what later became known as the RSPCA - Christopher Otter, “Cleansing and Clarifying: Technology and Perception in Nineteenth[1]Century London”, (2004) 43 (1) Journal of British Studies, 40, 45

 Also UK not the first and oldest society to enact anti-cruelty laws – eg  in the seventeenth century the following law and  were enacted:

1635, Act Against Plowing by the Tayle, and pulling the Wooll off Living Sheep 1635 (Ireland), 901 Statutes Passed in the Parliaments Held in Ireland, Vol 1 From the Third Year of Edward the 902 Second, A.D. 1310, to the Fourteenth and Fifteenth Years of Charles the Second, A.D. 1662 903 inclusive, Printed by George Grierson, Dublin (1794), 301.

Statutes Passed in the Parliaments Held in Ireland: 1310-1662 - Ireland - Google Books

The Massachusetts Body of Liberties 1641. https://history.hanover.edu/texts/masslib.html

Author Response

This is a very interesting manuscript which readers will find useful.

 Line 29 – perhaps it could be clearer whether the absence of standards is found in the UK, USA, EU (or all of them?)

RESPONSE: The sentence has been clarified

Line 36 – to whom does “their” refer –  perhaps instead, say something along the lines that that animals have coexisted with humans from prehistoric times – you can use this is a reference  Linda Kalof, Looking at Animals in Human History, Reaktion Books, London (2007) 8-10.

RESPONSE: The reference has been included in the text and in the bibliography, changing the numbering

Lines 9 and 39 – what is the seventh art.   Sorry for showing my ignorance!  I googled it and found out. There might be other readers who are also not aware but who are very interested in the subject matter of this article.

RESPONSE: The sentence has been changed

Lines 89-90 refer to “some European laws and treaties”  - would be helpful to cite a few of these.

RESPONSE: The sentence has been clarified

Line 95 – is this study summarised in Table 1 – perhaps let the reader know.

RESPONSE: it has been included in the text

Lines 97-101 – this is up to the editors – but the paragraph refers to websites and it would be helpful to the reader if these were cited.

Line 104 – “animal sacrifice” – could this be clarified. To English readers this invariably means some type of religious ritual where animals are killed.

RESPONSE: it has been changed

Line 105 “at that time” is a little unclear because previous sentences refer to 1957 and 1974 – perhaps it could be replaced with “from the late 1940s” – or something similar.

RESPONSE: it has been changed

Line 115 about the EU and its animal standards – I agree but it is such a broad statement a reference would be useful.   Eg https://www.fondation-droit-animal.org/proceedings-aw/the-european-union-legislation-on-animal-welfare/

RESPONSE: it has been included

Line 149 – perhaps point out that this declaration is not binding.

RESPONSE: It has been changed, pointing out that this declaration is not binding.

Line 159 – is there a reference for the conclusion that the article is rarely used – or is this conclusion drawn from the researchers’ own work.

RESPONSE: it has been changed

Line 164 “each of the EU countries” – appears that not all EU countries have specific laws on filming – perhaps this wording can be amended as it appears to say that all EU countries have laws with respect to filming (made clear in line 166 onwards).

RESPONSE: it has been changed

Line 178 – would be helpful to cite the Latvian law https://likumi.lv/ta/en/en/id/14940-animal-protection-law

RESPONSE: it has been included

Lines 206-7 – interesting comment regarding wild fauna.  Is this omission, however deliberate so that “pest” wild fauna are not subject to the law -  are there “environmental” type laws that would cover filming in natural spaces?  In New South Wales eg we have the Filming Approval Act 2004 (NSW) http://www.austlii.edu.au/cgi-bin/viewdb/au/legis/nsw/consol_act/faa2004177/  -   this is not designed to protect animal wellbeing but to protect the environment, which includes native animals, insects, etc.      As an aside we also have a code of practice specifically dealing with animals used in films (the code is mandatory under the Prevention of Cruelty to Animals Act 1979 (NSW) ) https://www.dpi.nsw.gov.au/animals-and-livestock/animal-welfare/animal-care-and-welfare/livestock/animal-use/animals-in-film-and-theatrical-performances/animals-in-film-and-theatrical-performances/code-of-practice-for-the-welfare-of-animals-in-films-and-theatrical-performances-plain-english

Line 213 – very interesting!

Line 230 – what is meant by “autonomous communities” – do you mean different stakeholder groups or local government (not sure)

RESPONSE: It has been clarified

Line 239 “extension” – do you mean given the breadth of the subject

RESPONSE: It has been clarified

Line 276 onwards – this part of the analysis requires modification – need to explain that USA and UK have different political systems which means they have different ways of dividing  legislative powers and competencies.

RESPONSE: It has been clarified

Lines 282-and generally that parag. USA has a similar federal system to Australia. Animal welfare laws are largely left up to the state legislatures and the federal government only becomes involved when it is considered a federal matter eg international trade, border controls for movement of animals and their products for biosecurity reasons.

RESPONSE: It has been clarified

There are no federal laws in USA dealing with films and animals – guidelines exist – I do not know I these are binding  (OK you refer to thse from line 318 onwards)

https://www.americanhumane.org/app/uploads/2016/08/Guidelines2015-WEB-Revised-110315-1.pdf

Also have a look at this https://www.animallaw.info/article/overview-laws-concerning-animals-film-media#:~:text=In%20response%2C%20a%20select%20few,a%20film%20depicting%20animal%20cruelty.   “Unfortunately, such anti-cruelty laws are ineffective against those depicting animal cruelty in films since it is difficult to identify the individual in the film. In response, a select few states specifically criminalize the filming of animal cruelty including California, Illinois, and Maine. These laws prohibit individuals from knowingly creating, selling, marketing, possessing a film depicting animal cruelty.”

The UK has general laws that can apply to Britain, Wales and Northern Ireland. Scotland has some legislative powers devolved on it.

See also Wild Animals in Circuses Act 2019 – for exhibited animals https://www.legislation.gov.uk/ukpga/2019/24/contents/enacted   (does not apply to all animals)

 Lines 308-309 referring to anti-cruelty laws and RSPCA – needs modification.  The 1822 Act applied to farm animals such as cattle – not all animals eg it applied to horses, cattle and sheep and did not deal with welfare in the sense the term evolved in the second part of the 20th century. Rather, it deal with what can be described as abject cruelty.

The following article contains material that details how the failure to implement the 1822 Act led to the creation of what later became known as the RSPCA - Christopher Otter, “Cleansing and Clarifying: Technology and Perception in Nineteenth[1]Century London”, (2004) 43 (1) Journal of British Studies, 40, 45

 Also UK not the first and oldest society to enact anti-cruelty laws – eg  in the seventeenth century the following law and  were enacted:

1635, Act Against Plowing by the Tayle, and pulling the Wooll off Living Sheep 1635 (Ireland), 901 Statutes Passed in the Parliaments Held in Ireland, Vol 1 From the Third Year of Edward the 902 Second, A.D. 1310, to the Fourteenth and Fifteenth Years of Charles the Second, A.D. 1662 903 inclusive, Printed by George Grierson, Dublin (1794), 301.

Statutes Passed in the Parliaments Held in Ireland: 1310-1662 - Ireland - Google Books

The Massachusetts Body of Liberties 1641. https://history.hanover.edu/texts/masslib.html

RESPONSE: All comments have been taken into account to modify this final part of the paper

Reviewer 2 Report

I see some merit in raising the issue of legal protection of animals used in production of films.

However I see a number of shortcomings of the current manuscript:

Firstly, there is very little attempt to explain how legislation in this field could and should work. Some more specific suggestions about what sensible legal requirements could be and through which instruments they they should be applied (e.g. by setting up a system where a committee should give permission like with use of animals for experimentation).

Secondly, the presentation of European legislation is problematic in a number of respects:

a) The very important distinction between EU law and treatises within the Council or Europe is not really explained.

b) Regulations and directives which are very different instruments within
the EU legislation are not clearly distinguished, and the nature of Directives
is not explained.

c) The Universal Declaration of Animal Rights is presented as if it is part of
European legislation, which it is not.

Thirdly, the very detailed presentation of local Spanish codes can only be of limited
interest in an international context.

Fourtly, when it comes to presentation of legislation from the US and the UK the
authors fail to acknowledge that these countries have a very different legal system
from that found in Spain and most other EU countries, i.e. common law. And this
has consequences for the role of various codes like those presented.

Finally the text is full of sentences that make very little sense, e.g.:
"Law 32/2007 on the care of animals, during farming, transport, experimentation, and slaughter [53], considers it a very serious

offense "to use animals in cinematographic, television, artistic or advertising productions, even with the permission of the authority, when the animal death"." (l. 225-228)

"Providers who have

recent and/or repeated incidents of animal abuse or neglect or other USDA violations related to the care and treatment of animals." (l. 326-328

Author Response

I see some merit in raising the issue of legal protection of animals used in production of films.

However I see a number of shortcomings of the current manuscript:

Firstly, there is very little attempt to explain how legislation in this field could and should work. Some more specific suggestions about what sensible legal requirements could be and through which instruments they they should be applied (e.g. by setting up a system where a committee should give permission like with use of animals for experimentation).

Secondly, the presentation of European legislation is problematic in a number of respects:

  1. a) The very important distinction between EU law and treatises within the Council or Europe is not really explained.

RESPONSE: It has been clarified in the text

  1. b) Regulations and directives which are very different instruments within
    the EU legislation are not clearly distinguished, and the nature of Directives
    is not explained.

RESPONSE: It has been clarified in the text

c) The Universal Declaration of Animal Rights is presented as if it is part of
European legislation, which it is not.

RESPONSE: It has been clarified, specifying that the Universal Declaration of Animal Rights is not binding.

Thirdly, the very detailed presentation of local Spanish codes can only be of limited
interest in an international context.

RESPONSE: We understand the question, but the paper tries to highlight the scarce existing regulations for animals in films and the great variation depending on the country or even the region, using Spain as a reference.

Fourtly, when it comes to presentation of legislation from the US and the UK the
authors fail to acknowledge that these countries have a very different legal system
from that found in Spain and most other EU countries, i.e. common law. And this
has consequences for the role of various codes like those presented.

RESPONSE: It has been clarified in the text

Finally the text is full of sentences that make very little sense, e.g.:
"Law 32/2007 on the care of animals, during farming, transport, experimentation, and slaughter [53], considers it a very serious offense "to use animals in cinematographic, television, artistic or advertising productions, even with the permission of the authority, when the animal death"." (l. 225-228)

 RESPONSE: It has been changed in the text

"Providers who have recent and/or repeated incidents of animal abuse or neglect or other USDA violations related to the care and treatment of animals." (l. 326-328

RESPONSE: It has been changed in the text

Reviewer 3 Report

This is a confused paper that should contain useful information on what standards there are on filming animals in different countries.  The paper suffers from poor English but does not set its goal of whether to review standards and legislation in countries on use of animals in film or appears at the end to argue that the EU has competence to legislate in this area.  It does not.  The Article 13 that is referred to does not give a legal base for EU law. The paper previous to this did not discuss if the EU did or did not have competence in this area but gave a number of useful indicators to what the different national laws are on use of animals in filming.  

There is a useful paper in here comparing different standards.  But it needs rewriting to focus on that area alone.

line 104 sacrifice should be changed to slaughter - 

Table 1 there are 44 different EU derived animal welfare laws and this Table only contains 8 so this should be improved or mentin made of the others

line 121 - there is a difference between Regulations which the MS cannot change and Directives which they can change - this needs to be clearer in the text

line 127 - list the countries that have ratified the Convention on pet animals 

line 127 there are also Conventions on slaughter and transport of farm animals and as the Regulations are mentioned in Table 1 these shoudl also be mentioned here 

line 197 as Spain is a Federal country and animal welfare laws are done at the State level rather than the Federal level this needs to be explicitly mentioned here 

Table 3 it is not clear if this Table is Federal legislation derived from EU Regulations and so directly applicable to all Spain or State legislation which would be different and derived from EU Directives; as there is no EU law on animal cruelty, as this is a issue devolved to MSs and then at the Spanish level devolved again to States and  Autonomous Regions

line 231 there is no EU derived law on protection of pets other than commercial and non commercial trade so its not clear what this line is referring to

line 283 - there are two Federal laws in the US - one on transport and the other on slaughter; the Animal Welfare Act is not a generic animal welfare law as this is devolved to States. The Animal Welfare Act brings in standards on use of animals in laboratories (but excludes mice and rats) and on sale of animals but is widely regarded as very weak and is not enforced 

line 293 the Animal Welfare Act 2006 prohibits any cruelty or suffering to animals used in films and videos if done on UK ground even if it does not specifically mention films

line 326 American Humane are the only organisation that goes on sets of films to authorise standards when using animals in film - the USA, despite having the weakest laws on filming have the highest audit system especially in Hollywood where AH would be on set to see their non statutory standards are upheld - they are then credited at the end of films with this role 

line 345 Article 13 of the TFEU has an exemption for cultural and religious issues so for instance does not cover bull fighting

line 351 there is no legal basis for EU standards on filming as it is not within EU competence as it is not a trade issue

line 352 I am not sure if the legal basis being suggested is a EU legal basis or a national legal base - if the latter it is correct to argue this, if the former it is not correct.

Author Response

This is a confused paper that should contain useful information on what standards there are on filming animals in different countries.  The paper suffers from poor English but does not set its goal of whether to review standards and legislation in countries on use of animals in film or appears at the end to argue that the EU has competence to legislate in this area.  It does not.  The Article 13 that is referred to does not give a legal base for EU law. The paper previous to this did not discuss if the EU did or did not have competence in this area but gave a number of useful indicators to what the different national laws are on use of animals in filming. 

RESPONSE:  The paper has been corrected in English and all suggested comments have been corrected. Article 13 of the Treaty on the Functioning of the European Union says "In formulating and implementing the Union's agriculture, fisheries, transport, internal market, research and technological development and space policies, the Union and the Member States shall, since animals are sentient beings, pay full regard to the welfare requirements of animals, while respecting the legislative or administrative provisions and customs of the Member States relating in particular to religious rites, cultural traditions and regional heritage". Thus, any new legislation or modification of the existing one to consider animals as sentient beings and the treaty is mandatory, since it is signed by all the heads of state of the member countries. Thus animal welfare as sensitive-sentient beings, there is no obligation for the European legislator, nor for the Member States, to ensure its welfare, but there is an obligation to prevent regulations that may affect that welfare from totally ignoring it both in the moment of making the decision (procedural aspect) and the result (substantive aspect).

There is a useful paper in here comparing different standards.  But it needs rewriting to focus on that area alone.

line 104 sacrifice should be changed to slaughter – 

RESPONSE: The sentence has been changed

Table 1 there are 44 different EU derived animal welfare laws and this Table only contains 8 so this should be improved or mentin made of the others

RESPONSE: It has been corrected

line 121 - there is a difference between Regulations which the MS cannot change and Directives which they can change - this needs to be clearer in the text

RESPONSE: It has been changed and corrected in the text

line 127 - list the countries that have ratified the Convention on pet animals

RESPONSE:  A sentence has been included referring to the countries that have ratified the Convention on pet animals, and we have put the reference where the list appears. It seemed to us that listing the 26 countries would greatly lengthen the paper.

line 127 there are also Conventions on slaughter and transport of farm animals and as the Regulations are mentioned in Table 1 these shoudl also be mentioned here 

RESPONSE:  It has been included

line 197 as Spain is a Federal country and animal welfare laws are done at the State level rather than the Federal level this needs to be explicitly mentioned here

RESPONSE: It has been clarified. Spain is not a federal state, but a state of autonomies, where they have many powers but not all. Thus, the state sets guidelines and later the autonomous communities that have this transferred power can implement the standard in their own autonomy or stay with the general state standard. The rules that are derived from the autonomous communities will always be the same or more restrictive but never less than what is established by the state standard.

Table 3 it is not clear if this Table is Federal legislation derived from EU Regulations and so directly applicable to all Spain or State legislation which would be different and derived from EU Directives; as there is no EU law on animal cruelty, as this is a issue devolved to MSs and then at the Spanish level devolved again to States and  Autonomous Regions

RESPONSE: It has been clarified in the text

line 231 there is no EU derived law on protection of pets other than commercial and non commercial trade so its not clear what this line is referring to

RESPONSE: It has been clarified in the text

line 283 - there are two Federal laws in the US - one on transport and the other on slaughter; the Animal Welfare Act is not a generic animal welfare law as this is devolved to States. The Animal Welfare Act brings in standards on use of animals in laboratories (but excludes mice and rats) and on sale of animals but is widely regarded as very weak and is not enforced 

RESPONSE: it has been included

line 293 the Animal Welfare Act 2006 prohibits any cruelty or suffering to animals used in films and videos if done on UK ground even if it does not specifically mention films

RESPONSE: it has been included

line 326 American Humane are the only organisation that goes on sets of films to authorise standards when using animals in film - the USA, despite having the weakest laws on filming have the highest audit system especially in Hollywood where AH would be on set to see their non statutory standards are upheld - they are then credited at the end of films with this role 

RESPONSE: it has been included

line 345 Article 13 of the TFEU has an exemption for cultural and religious issues so for instance does not cover bull fighting

RESPONSE: it has been included

line 351 there is no legal basis for EU standards on filming as it is not within EU competence as it is not a trade issue

RESPONSE: it has been corrected

line 352 I am not sure if the legal basis being suggested is a EU legal basis or a national legal base - if the latter it is correct to argue this, if the former it is not correct.

RESPONSE: it has been corrected

Round 2

Reviewer 3 Report

Lin 414 the Council of Europe Convention is not a European law - there is no obligation to ratify or follow it's terms and it is separate to the the acquis. 

Line there is no legal base in the  Treaty of Rome for animal welfare yherefore any legislation must be justified under a specific Article and so legal base this is usually environment or trade.  This paper does not state that.  It is therefore impossible for the EU to enact legislation on standards in filming unless it is for trade purposes such as the data protection legislation.

The conclusion doesn't fit the previous discussion - the authors dont explain the limitsnof the treaty of Rome in being he legal base for animal welfare laws - unless there are trade issues (and there aren't here unless for data protection) you cannot have EU standards in animals in filming - - it would fall under the subsidiarity principles and be left to Member states

Author Response

Lin 414 the Council of Europe Convention is not a European law - there is no obligation to ratify or follow it's terms and it is separate to the the acquis. 

RESPONSE: Paragraph has been changed and clarified according suggestion of review.

Line there is no legal base in the  Treaty of Rome for animal welfare Therefore any legislation must be justified under a specific Article and so legal base this is usually environment or trade.  This paper does not state that.  It is therefore impossible for the EU to enact legislation on standards in filming unless it is for trade purposes such as the data protection legislation.

RESPONSE: A sentence has been inserted in the text to clarify this point regarding the Treaty of Rome.

The conclusion doesn't fit the previous discussion - the authors dont explain the limitsnof the treaty of Rome in being he legal base for animal welfare laws - unless there are trade issues (and there aren't here unless for data protection) you cannot have EU standards in animals in filming - - it would fall under the subsidiarity principles and be left to Member states

RESPONSE: The conclusion has been changed to clarify that the TFEU is not a legal basis and the issue of animal welfare is not an objective of the European